# Non-isocyanate polyurethanes synthesized from terpenes using thiourea organocatalysis and thiol-ene-chemistry

Frieda Clara M. Scheelje[1] & Michael A. R. Meier [1,2]✉

The depletion of fossil resources as well as environmental concerns contribute to an increasing focus on finding more sustainable approaches for the synthesis of polymeric materials. In this work, a synthesis route towards non-isocyanate polyurethanes (NIPUs) using renewable starting materials is presented. Based on the terpenes limonene and carvone as renewable resources, five-membered cyclic carbonates are synthesized and ring-opened with allylamine, using thiourea compounds as benign and efficient organocatalysts. Thus, five renewable AA monomers are obtained, bearing one or two urethane units. Taking advantage of the terminal double bonds of these AA monomers, step-growth thiol-ene polymerization is performed using different dithiols, to yield NIPUs with molecular weights of above 10 kDa under mild conditions. Variation of the dithiol and amine leads to polymers with different properties, with $M_n$ of up to 31 kDa and $T_g$'s ranging from 1 to 29 °C.

[1] Laboratory of Applied Chemistry, Institute of Organic Chemistry (IOC), Karlsruhe Institute of Technology (KIT), Straße am Forum 7, 76131 Karlsruhe, Germany. [2] Laboratory of Applied Chemistry, Institute of Biological and Chemical Systems - Functional Molecular Systems (IBCS-FMS), Karlsruhe Institute of Technology (KIT), Hermann-von-Helmholtz-Platz 1, 76344 Eggenstein-Leopoldshafen, Germany. ✉email: m.a.r.meier@kit.edu

With a current global market volume of 25 million tons[1], polyurethanes are an indispensable class of polymers with versatile applications in our daily life, such as coatings, foams and adhesives[2]. With regard to several hazards related to their production, including the use of toxic phosgene[3] for the synthesis of isocyanates, which are hazardous themselves[4], scientific effort within the last years has been put increasingly into the development of isocyanate-free routes to polyurethanes[5–9]. One possibility to access these non-isocyanate polyurethanes (NIPUs) is the ring-opening of cyclic carbonates with amines, as was reviewed extensively[10,11].

Apart from the development towards less hazardous synthesis routes, the use of renewable feedstock for the production of polymeric materials, including polyurethanes, is highly desirable with regard to overall sustainable procedures[12–14]. Suitable monomers can be obtained from different renewable resources, such as oleochemicals[15,16], tannin[17] or terpenes[18–20]. As an example, cyclic carbonates derived from epoxidized soybean oil have been used for the synthesis of a variety of polymeric materials containing urethane units[21–23].

Due to their structural diversity and their occurrence as waste products in the chemical industry[24–26], terpenes have gained increasing research interest for the production of fine chemicals[27], pharmaceutical products[28] and polymeric materials[29–35]. In works by the group of Mülhaupt[36,37] as well as by Della Monica and Kleij et al.[38], the use of terpene-derived dicarbonates as AA monomers for the step-growth synthesis of NIPUs was investigated. However, in all cases, oligomers with limited molar masses were obtained due to viscosity reasons, thus requiring further reaction steps for the use of these pre-polymers.

The ring-opening of cyclic carbonates with amines can be catalyzed by the addition of Lewis acids[39] and also by organo-catalysts such as 1,5,7-triazabicyclo[4.4.0]dec-5-ene[40–42], which was also already applied in polymer synthesis[43]. Another possible compound group that can be used for the aminolysis of cyclic carbonates are thioureas[40,42,44,45], which coordinate to carbonyl groups[46] and thus can activate cyclic carbonates via hydrogen bonding[47,48]. For efficient hydrogen bonding, electron-withdrawing substituents are beneficial, further, aromatic groups can increase the activation by preorganization[49,50]. As such, the 3,5-bis(trifluoromethyl) phenyl group is often used in thiourea catalysts[50], but can also be substituted by other electron-withdrawing groups[51–55]. With the aim to develop more sustainable processes, it is possible to synthesize such thiourea compounds via a multicomponent reaction from an isocyanide, an amine and base-activated sulfur[55–57]. With this, thiourea organocatalysts are possibly attractive with respect to the generation of urethane moieties within polyurethanes.

A possibility to obtain polyurethanes is by already implementing the urethane motif into a monomer[58]. In previous work of our group[59], polyurethanes were synthesized not by polyaddition of amines and cyclic carbonates, but by synthesizing monomers that contained a urethane functional group and two terminal double bonds. The urethane moieties were obtained via a Lossen rearrangement[60] starting from fatty acid-based hydroxamic acids. The terminal double bonds could then react with renewable dithiols in a step-growth polymerization via thiol-ene reaction. This enabled the accessibility of renewable polyurethanes with high molar masses under comparably mild conditions.

Extending this approach, we synthesized different renewable urethane monomers from terpenes, bearing terminal double bonds for polymerization using a thiol-ene approach. To this end, we made use of previously described cyclic carbonates from terpenes in order to open them with amines containing a double bond. For the aminolysis, the effect and possible use of thiourea catalysts was investigated thoroughly.

## Results

**Synthesis of carbonate monomers.** The epoxides **3-5** and thereof derived carbonates **6-8**, respectively derived from limonene **1** and carvone **2**, were synthesized on the basis of previous studies[61–64]. With limonene containing two double bonds, it is possible to oxidize either both double bonds[62] using common epoxidizing agents or to oxidize the higher substituted double bond selectively by reaction with *N*-bromo succinimide (NBS) and sodium hydroxide[61]. Also in the case of carvone **2**, due to the additional keto group when compared to limonene, the two double bonds show different reactivity, and it is possible to address them selectively[62].

Based on this, the respective cyclic carbonates **6-8** were synthesized from the respective epoxides by insertion of $CO_2$ using tetrabutylammonium chloride (TBACl) as catalyst[63]. Despite recent findings showing high yields for the formation of cyclic carbonates when using aluminum catalysts[63], it was decided to constrain the use of catalytic substances to the halogenide for a simpler approach. Indeed, no additional catalyst was necessary for a sufficient conversion of the epoxides. The synthesis of the cyclic carbonates is shown in Fig. 1. In the case of the two double bonds of carvone **2**, only compound **8** bearing an exocyclic carbonate could be isolated selectively.

It must be noted that the reactivity of the endocyclic and exocyclic terpene epoxides varies depending on the substitution pattern. In limonene dioxide **4**, this could be observed in the formation of a monocarbonate intermediate **7a**, in which the epoxide attached to the ring was not yet converted into the carbonate (see Fig. 2).

By isolating the intermediate **7a**, the ratio of **7** to **7a** could be determined and compared with respect to different reaction conditions (see Supplementary Table 1). As catalysts for the carbonate formation, tetrabutylammonium chloride (Supplementary Table 1, entries 1 and 4), bromide (entry 2) and iodide (entry 3) were used and the ratio of **7** to the starting material **4** and the

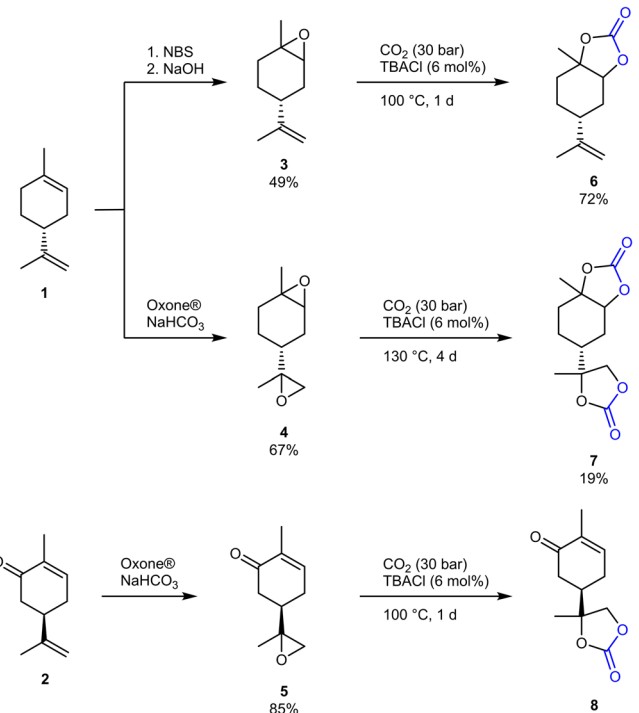

**Fig. 1 Synthesis of cyclic carbonates from limonene and carvone.** After selective oxidation of the terpene double bonds, epoxides **3–5** are converted into the respective cyclic carbonates by $CO_2$ insertion.

intermediate product **7a** was compared using GC-FID measurements. It was shown that, in order to obtain full conversion of the limonene dioxide **4**, stirring at 130 °C for three days with TBACl as catalyst was necessary. This low reactivity of the internal epoxide was not observed in the case of **3**, a difference that could be explained by a lower reactivity of **4** as well as by an increase in viscosity after the carbonation of one epoxide group in **7a**, thus making longer stirring and higher temperature necessary for further carbonation towards the final product **7**.

When using NBS and sodium hydroxide for the oxidation of limonene, one diastereomer of **3** is formed selectively, fitting to previous findings that the *trans* isomer was mainly observed[61]. The diastereomeric ratio was determined to be 93:7 by GC-FID and NMR spectroscopy (see Supplementary Figs. 19 and 20). During the carbonate formation, the stereocenter is retained, thus also in compound **6** one main diastereomer is observed (see Supplementary Figs. 23 and 24). On the other hand, oxidation of carvone to the epoxide **5** led to the formation of two diastereomers in nearly equimolar ratio (see Supplementary Figs. 37 and 38), which was also preserved in the carbonate formation (see Supplementary Figs. 41 and 42). For the formation of **7**, a commercially available diastereomeric mixture of limonene dioxide **4** was used, and in compound **7**, two isomers were observed by NMR spectroscopy in a ratio of 51:49 (see Supplementary Fig. 34).

### Ring-opening of carbonates with amine

*Opening of endocyclic carbonate group.* To test if thiourea catalysts can effectively promote the ring opening of cyclic carbonates **6–8** with an amine, four different thiourea catalysts **9–12** were compared with respect to their capability to activate the cyclic carbonate (see Fig. 3). All of these catalysts are accessible via a multicomponent approach, which is significantly more sustainable than classic synthesis approaches, from the respective aromatic isocyanide, elemental sulfur and cyclohexyl amine. Depending on the residue on the aromatic ring, they vary in their hydrogen bonding ability, with catalysts **9** and **10** showing the strongest affinity towards carbonyl groups due to the presence of strong electron-withdrawing groups[55].

Opening of the cyclic carbonates described above with amines containing a double bond leads to monomers containing a urethane moiety and two terminal double bonds, thus making them suitable building blocks for potential NIPU synthesis.

For the investigation of a possible catalytic effect upon opening the synthesized carbonates with amines, limonene monocarbonate **6** was chosen as model compound to analyze the opening to the endocyclic carbonate adjacent to the terpene ring structure separately. As nucleophile, allylamine was used. Investigation started using thiourea **9**, as it is easily accessible from the commercially available isothiocyanate and was shown to be a promising compound for the activation of carbonyl compounds[50].

As in the synthesis of the carbonate **6** a high conversion was achieved, the crude reaction mixture could directly be used for the urethane synthesis after washing with brine, without the need of a purification via column chromatography in between. Further, no solvent was necessary for both the carbonate and the urethane formation, thus contributing to the overall sustainability of the synthesis. The ratio of the urethane monomer **13** to the carbonate **6** was determined via GC-FID over time, as no other signals were observed, and is shown in Fig. 4 (for exact values, see Supplementary Table 2).

It can be stated that the reaction is significantly more effective in the presence of thiourea **9**, indicating a catalytic effect of the thiourea that positively influences the reaction efficiency. During purification of the urethane monomer **13**, the thiourea **9** could be recovered, corresponding to 100% of the starting amount, thus further contributing to the sustainability of the presented approach. The presence of a high percentage of intact thiourea in the final reaction mixture indicates that the thiourea compound is indeed acting as a catalyst. The monomer **13** could be isolated in a yield of 85%.

The unsymmetric carbonate **6** can be opened in two ways, thus two regioisomers are possibly formed during the reaction. 2D NMR spectra confirm the presence of two regioisomers (see Supplementary Figs. 27 and 28). From GC-FID measurements, their ratio was determined to be 53:47, indicating low regioselectivity. Thus, only one of the two regioisomers of **13** is shown in Fig. 4 for clarity.

*Variation of reaction conditions.* To optimize the reaction conditions, equivalents and temperature were varied (see Table 1, entries 1–5). Further, alternative thiourea compounds **10–12** were tested next to compound **9**, displaying a range of stronger and weaker electron-withdrawing groups (entries 6–8, Table 1).

As can be seen from entry 2 (Table 1), it is possible to reduce the amount of catalyst used, but higher yields were obtained for 5 mol%, thus further experiments were carried out using this catalyst loading. Reduction to only one equivalent of allylamine led to a lower conversion (entry 3, Table 1), probably due to the volatility of the amine. Further, the initial choice of 70 °C (entry 1, Table 1) proved to be the most suited for the investigated reaction if compared to temperatures of 60 and 80 °C (see entries 4 and 5, Table 1).

The use of alternative catalysts **10–12** also led to the formation of product, with electron withdrawing sulfone (entry 6, Table 1) and ester groups (entry 7, Table 1) resulting in higher conversions than in the case of a simple phenyl residue (entry 8, Table 1). However, catalyst **9** bearing two $CF_3$ groups still showed the highest activity. This confirmed the choice of catalyst **9** for further

**Fig. 2 Carbonation of limonene dioxide 4.** Formation of dicarbonate **7** and monocarbonate **7a** were observed by gas chromatography.

**Fig. 3 Thiourea organocatalysts used within this work.** The choice of substituents determines the hydrogen bonding ability of these thiourea catalysts.

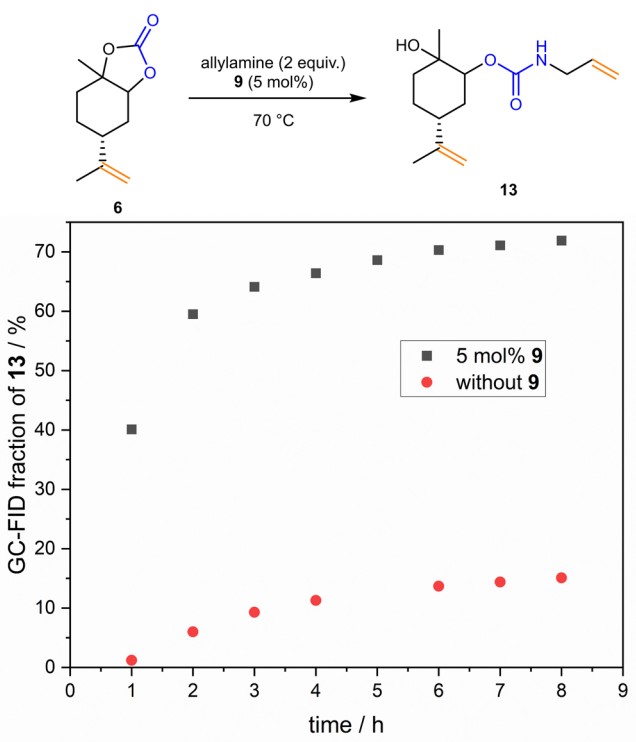

**Fig. 4 Conversion of limonene-derived carbonate 6 to urethane monomer 13.** The GC-FID fraction of **13** is obtained by dividing the GC integral of the signal associated with **13** by the sum of the integrals of the signals assigned to **6** and **13**. Only one of two formed regioisomers of **13** is shown for clarity.

**Table 1 Variation of thiourea concentration, reaction temperature, and choice of thiourea catalysts in the carbonate ring-opening of 6.**

| Entry | Thiourea | Mol% thiourea | Equiv. amine | T/°C | 13:6 (4 h) | 13:6 (8 h) |
|---|---|---|---|---|---|---|
| 1 | **9** | 5.0 | 2.0 | 70 | 61:39 | 76:24 |
| 2 | **9** | 2.5 | 2.0 | 70 | 45:55 | 57:43 |
| 3 | **9** | 5.0 | 1.0 | 70 | 27:73 | 37:63 |
| 4 | **9** | 5.0 | 2.0 | 60 | 55:45 | 68:32 |
| 5 | **9** | 5.0 | 2.0 | 80 | 58:42 | 66:34 |
| 6 | **10** | 5.0 | 2.0 | 70 | 40:60 | 55:45 |
| 7 | **11** | 5.0 | 2.0 | 70 | 49:51 | 64:36 |
| 8 | **12** | 5.0 | 2.0 | 70 | 28:72 | 42:58 |

All reactions were carried out in sealed vials under bulk conditions. The GC-FID fraction of **13** is obtained by dividing the GC integral of the signal associated with **13** by the sum of the integrals of the signals assigned to **6** and **13**. Only one of two formed regioisomers of **13** is shown for clarity, with the ratio of **13:6** including both regioisomers.

experiments, yet catalysts **10** and **11** remain valid options that can be obtained more sustainably.

The investigations proved the initial reaction conditions from entry 1 (Table 1) to be the most suited for the opening of the endocyclic carbonate group. As this functionality is assumed to be

**Table 2 Carbonate ring-opening of 8 with and without thiourea catalyst.**

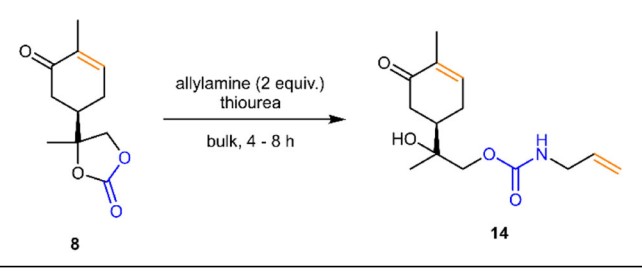

| Entry | Thiourea | Mol% thiourea | 14:8 (4 h) | 14:8 (8 h) |
|---|---|---|---|---|
| 1 | **9** | 5.0 | 83:17 | 90:10 |
| 2 | – | – | 78:22 | 96:4 |

All reactions were carried out in sealed vials under bulk conditions. The GC-FID fraction of **14** is obtained by dividing the GC integral of the signal associated with **14** by the sum of the integrals of the signals assigned to **8** and **14**. Only one of two formed regioisomers of **14** is shown for clarity, with the ratio of **14:8** including both regioisomers.

the less reactive one, the same conditions were chosen for a closer look into the opening in the exocyclic carbonate groups in the terpene derivatives **7** and **8**.

*Opening of exocyclic carbonate groups.* To analyze the influence of the thiourea catalyst on the exocyclic carbonate group found in compounds **7** and **8**, the carbonate **8** based on carvone was chosen as model substrate as it does not contain an additional endocyclic carbonate group.

For the opening of the carbonate, the optimized conditions of the opening of **6** (see Table 1, entry 1) were used. The conversion was detected via GC-FID as in the case before and the comparison of the reactions with and without the presence of thiourea **9** is shown in Table 2.

In general, higher conversions to the respective urethane **14** are observed, as expected, than in the case of the endocyclic carbonate (see Table 1), which can be attributed to a lower steric hindrance. Already without the addition of thiourea, high conversions are obtained (entry 2, Table 2). Addition of thiourea does not significantly influence the conversion (see entry 1, Table 2), thus no activation is necessary to obtain the urethane monomer **14** from carbonate **8**.

Similar to compound **6**, the carbonate **8** can in principle be opened on two different sides, leading to regioisomers. Based on NMR data (see Supplementary Figs. 43 and 44), mainly one regioisomer of **14** is formed. This can be related to the two sides of the carbonate group in **8** differing stronger in steric demand than in the case of **6**. Still, two different diastereomers are distinguishable via GC-FID and NMR measurements (see Supplementary Figs. 45 and 46).

Compound **7** bears two carbonate groups, which show different reactivities as demonstrated before. Therefore, thiourea catalyst **9** was used for the activation of especially the endocyclic carbonate. The general reaction conditions for the opening of **7** were chosen as in Table 1 (entry 1), using the double amount of allylamine and thiourea **9** (see Fig. 5).

As intermediate, the monourethane **15a** (see Fig. 5) could be isolated, confirming the observation that the endocyclic carbonate group is less reactive. The chemical similarity between the monourethane **15a** and the diurethane **15** results in a difficult separation via column chromatography, requiring a gradient column that hampers the recyclability of the solvent mixture. In order to facilitate the work-up, the reaction progress was monitored over time (see Supplementary Table 3), also

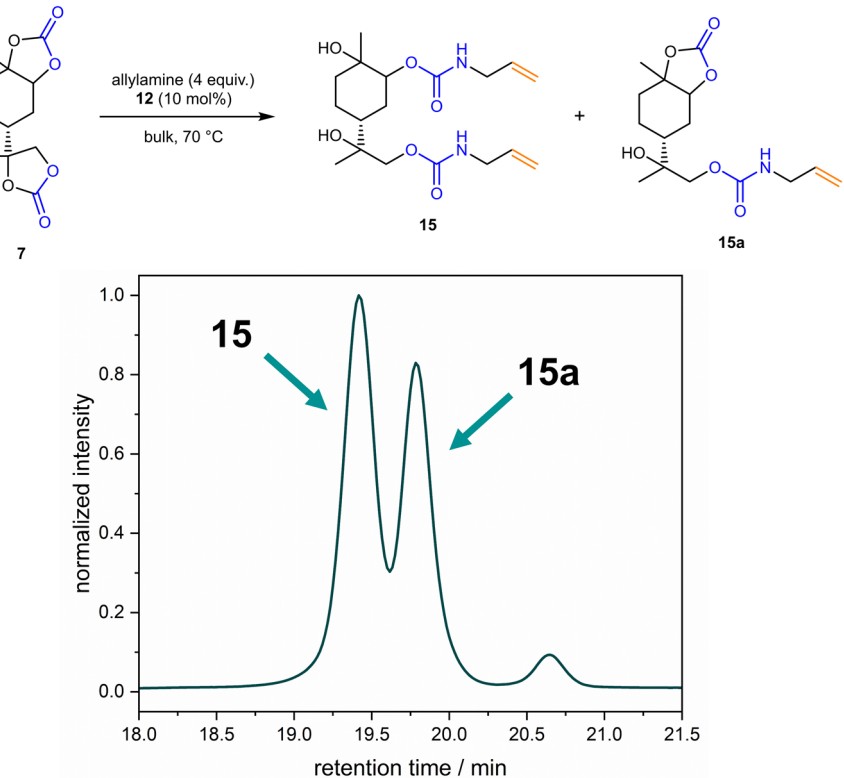

**Fig. 5 Ring-opening of limonene carbonate 7 to bifunctional urethane 15 and monofunctional urethane 15a, analyzed via SEC.** The SEC chromatogram was measured after 2 h reaction time, still showing presence of the intermediate **15a**. The signal at 20.7 min is a system peak and does not correspond to any compound found in the mixture. In the chemical equation, only one regioisomer of **15** is shown for clarity, respectively.

investigating whether it is possible to push the reaction to a quantitative conversion of **7** and **15a** by adding further allylamine after a certain amount of time. The low volatility of **15** required the use of SEC instead of GC-FID for monitoring, nevertheless enabling a qualitative observation of the presence of unreacted **15a** within the mixture. In entries 1b-7b (Supplementary Table 3), an additional equivalent of allylamine was added after 6 h. However, no difference in yield was observed compared to the batch where no allylamine was added at a later stage (see entries 1a–7a, Supplementary Table 3), and in both cases, full conversion of **15a** was not observed. The occurring of this structurally similar intermediate, which is not straightforward to isolate from the product, represents a drawback of the bifunctional monomer **15** when compared to monomer **13**. When stopping the reaction after one day, most of the intermediate was converted and after purification via column chromatography the product was isolated in 85% yield.

Regarding the regioselectivity of the reaction, an analogous trend was observed in the endocyclic and exocyclic carbonate groups as in compounds **13** and **14**. The exocyclic carbonate is opened selectively on the sterically less demanding side, whereas in the case of the endocyclic carbonate, both regioisomers are formed in a ratio of 45:55 as determined by NMR measurements (see Supplementary Fig. 34). Since compound **15** was not volatile enough to analyze the product mixture via GC-FID, the ratio can be determined with less accuracy due to overlapping signals in the NMR spectrum, however, the observations match the previous findings for **13** and **14**.

*Fatty acid-based amine.* Considering the toxicity of allylamine as well as solubility issues in the case of the substrate **15**, it is desirable to look for alternative amines containing a double bond. As such, fatty acid-based derivatives are desirable, being still simple in their structure. In previous work, it was shown that fatty acid-derived undecenoic acid can be used as starting material for the synthesis of decenyl amine (see Fig. 6) via esterification, substitution with hydroxyl amine, Lossen rearrangement and subsequent saponification[65]. The amine was thus synthesized according to the procedure described in the literature. The introduction of this moiety into the urethane monomers was first tried using limonene monocarbonate **6** (see Fig. 6). Analogous reaction conditions as in the case of allylamine were chosen (see Table 1, entry 1). Also in this case, two equivalents of the amine were used to enable higher conversion, and thiourea **9** was added to activate the endocyclic carbonate. The monomer **18** was obtained in a yield of 55%, the carbonate opening was thus less effective in comparison to the reaction with allylamine, which can be attributed to the lower reactivity of the amine due to its higher molecular weight[66].

Further, limonene dicarbonate **7** was opened with **17** using analogous reaction conditions, yielding the diurethane monomer **19** in a yield of 15%. The low yield can be attributed to both the carbonate and the amine being less reactive than their respective counterparts, as discussed above.

**Synthesis of linear NIPUs.** To demonstrate a possible application of the synthesized urethane monomers for polymer synthesis, the substrates **13–15**, **18**, and **19** were reacted with dithiols in a step-growth thiol-ene polymerization to obtain linear NIPUs. To promote a radical formation, 2,2-dimethoxy-2-phenylacetophenone (DMPA) was added as initiator and the reaction mixture was irradiated with UV light of 365 nm.

As promising substrate, limonene mono-urethane **13** was chosen as it showed good solubility in various solvents. Among commercially available dithiols, 1,10-decanedithiol **20** was chosen for first test reactions. It contains a linear spacer that is expected

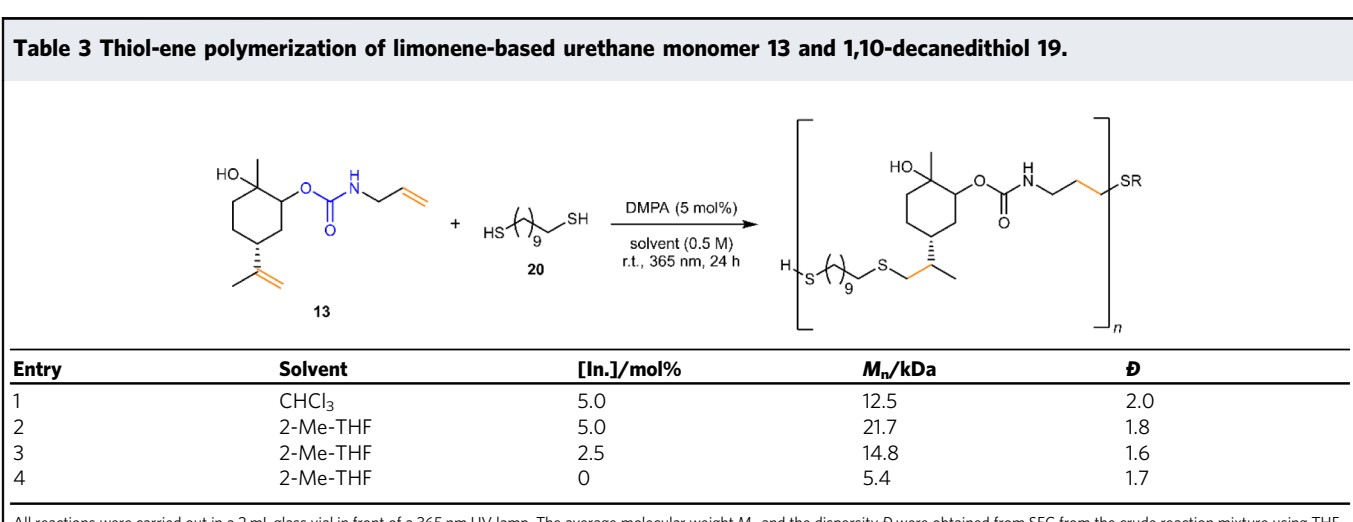

**Fig. 6 Synthesis of fatty acid-based amine 17 and subsequent carbonate opening of 6 and 7 to urethane monomers 18 and 19.** Only one of two formed regioisomers of **18** and **19** is shown for clarity, respectively.

**Table 3 Thiol-ene polymerization of limonene-based urethane monomer 13 and 1,10-decanedithiol 19.**

| Entry | Solvent | [In.]/mol% | $M_n$/kDa | Đ |
|---|---|---|---|---|
| 1 | CHCl$_3$ | 5.0 | 12.5 | 2.0 |
| 2 | 2-Me-THF | 5.0 | 21.7 | 1.8 |
| 3 | 2-Me-THF | 2.5 | 14.8 | 1.6 |
| 4 | 2-Me-THF | 0 | 5.4 | 1.7 |

All reactions were carried out in a 2 mL glass vial in front of a 365 nm UV lamp. The average molecular weight $M_n$ and the dispersity Đ were obtained from SEC from the crude reaction mixture using THF as solvent. Only one of two regioisomers of **13** is shown for clarity. In. = 2,2'-dimethoxy-2-phenylacetophenone.

to keep the polar urethane moieties at a sufficient distance to enable a certain degree of solubility. Test reactions were performed in chloroform, a solvent shown suitable in previous studies[59], and 2-methyl tetrahydrofuran (2-Me-THF), which represents a less hazardous and furthermore renewable solvent. A concentration of 0.5 M was found to be a good compromise between not using too much solvent and yet dissolving the monomers well enough to enable efficient stirring. The results of the polymerization reactions are shown in Table 3 and Supplementary Fig. 2, revealing that efficient formation of polymers with molecular weights >10 kDa takes place at the chosen conditions. Both solvents led to the formation of polymers (see Table 3, entries 1 and 2), with higher molecular weights being achieved in the case of the more sustainable 2-Me-THF.

To gain insight into the reaction taking place, control reactions were carried out in absence of either the diene monomer **13**, the dithiol **20**, UV irradiation or initiator. The results show that both monomers as well as irradiation with UV light are necessary for the formation of polymers. In absence of DMPA, oligomeric species with a molecular weight of $M_n = 5.4$ kDa were observed after two days of irradiation (see Table 3, entry 4), confirming that radical addition can also take place without an initiator. However, significantly higher molecular weights were achieved when adding DMPA to the reaction mixture. A reduction of the initiator concentration to 2.5 mol% also led to the formation of

polymers with $M_n > 10$ kDa (see Table 3, entry 3). Yet, this molecular weight is lower than when using 5 mol% initiator.

The polymer could be precipitated from the crude mixture by dropping the solution into cold methanol. The [1]H NMR spectrum of the precipitated polymer (see Supplementary Fig. 6) confirmed the presence of both the terpene moiety and the aliphatic chain of the dithiol within the material. Further, the IR spectrum of the precipitated polymer (see Supplementary Fig. 11) shows the presence of characteristic signals corresponding to the present urethane and thioether moieties. Together with the performed control reactions, this undermines the assumption of NIPU formation via thiol-ene polyaddition.

The formation of NIPUs from monomer **13** was monitored over time (see Supplementary Table 4 and Supplementary Fig. 1), showing that already after 1 h a molecular weight of 11.8 kDa was achieved. After 5 h, 14.1 kDa were observed, indicating that the polymerization can possibly be stopped earlier. Nevertheless, for a comparison between the different monomers, 24 h were kept as fixed reaction time to allow for slower reactions to still be observed.

After the first promising results, the determined reaction conditions were applied for the synthesis of linear NIPUs from all urethane monomers synthesized in this work. Besides the variation of the urethane monomer, a variation of the dithiol can be a possibility to achieve different properties. As renewable

**Table 4 Variation of diene and dithiol within the synthesis of NIPUs.**

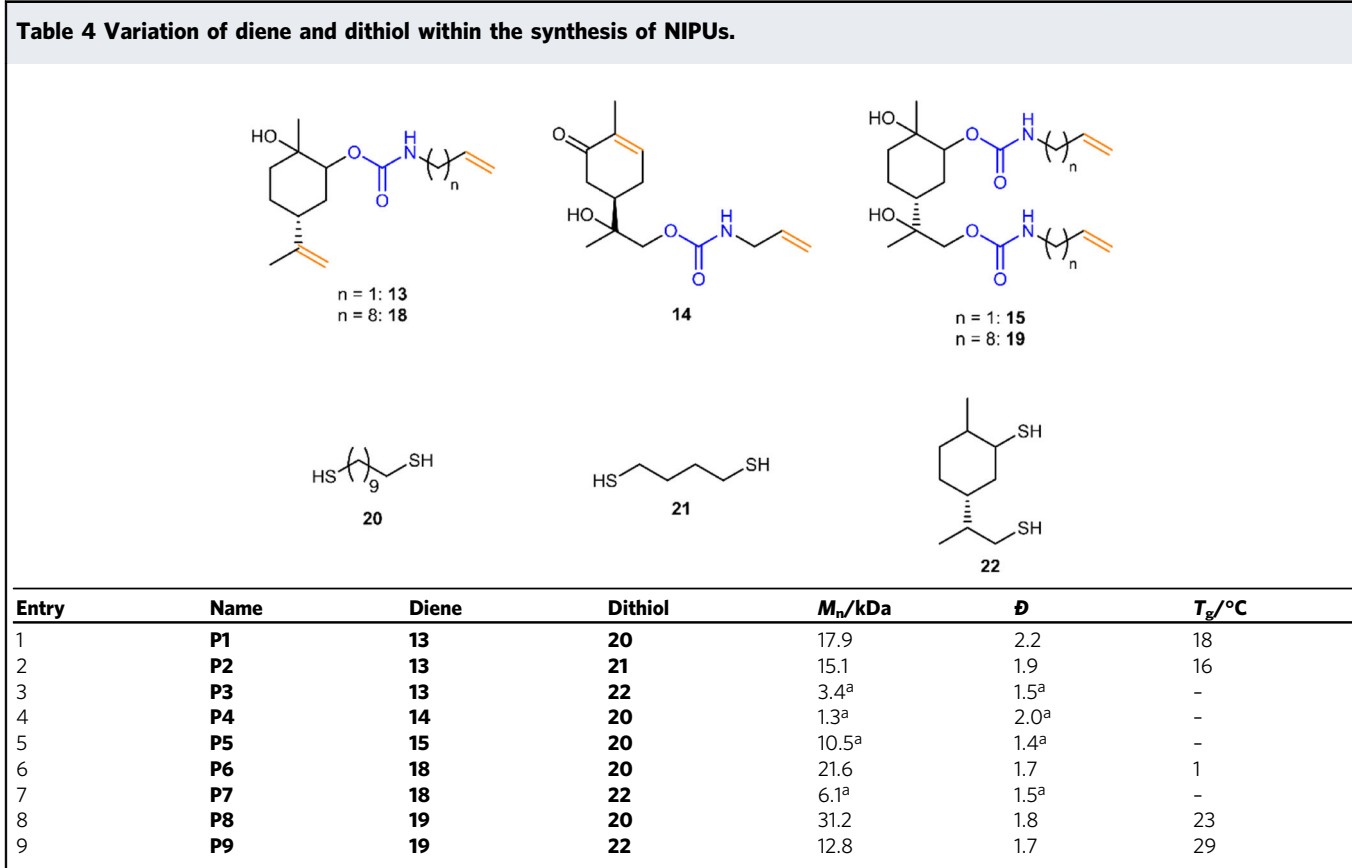

| Entry | Name | Diene | Dithiol | $M_n$/kDa | Đ | $T_g$/°C |
|---|---|---|---|---|---|---|
| 1 | **P1** | **13** | **20** | 17.9 | 2.2 | 18 |
| 2 | **P2** | **13** | **21** | 15.1 | 1.9 | 16 |
| 3 | **P3** | **13** | **22** | 3.4[a] | 1.5[a] | – |
| 4 | **P4** | **14** | **20** | 1.3[a] | 2.0[a] | – |
| 5 | **P5** | **15** | **20** | 10.5[a] | 1.4[a] | – |
| 6 | **P6** | **18** | **20** | 21.6 | 1.7 | 1 |
| 7 | **P7** | **18** | **22** | 6.1[a] | 1.5[a] | – |
| 8 | **P8** | **19** | **20** | 31.2 | 1.8 | 23 |
| 9 | **P9** | **19** | **22** | 12.8 | 1.7 | 29 |

All reactions were carried out in a 2 mL glass vial in front of a 365 nm UV lamp, with reaction conditions according to Table 3, entry 2. The average molecular weight $M_n$ and the dispersity Đ were obtained from SEC of the precipitated polymer using THF as solvent. The $T_g$ was obtained from DSC measurement of the dried polymer samples (Supplementary Fig. 16).
[a]Obtained from SEC of the crude reaction mixture, as precipitation of the polymer was not possible.

alternative to **20**, dithiol **22** was synthesized from limonene. A summary of the used dithiols and of the results from the polymerization reactions is shown in Table 4 and Supplementary Table 5. SEC of the obtained polymeric materials was performed (chromatograms are shown in Supplementary Figs. 3–5) to determine $M_n$ and Đ (see Table 4), revealing mostly high molecular weights and dispersities close to 2, as expected for a step-growth polymerization. In contrast to the results achieved with monomer **13**, the use of monomer **14** (entry 4, Table 4) only led to the formation of oligomeric species, which can be attributed to the reduced reactivity of the double bond in α,β-position to the carbonyl group with respect to radical thiol-ene additions. The urethane monomers **15**, **18** and **19** could successfully be used for the synthesis of NIPUs with $M_n$ > 10 kDa. When compared to monomer **13**, the polymer **P6** derived from monomer **18** shows slightly higher molecular weight (entry 6, Table 4). The resulting polymer was precipitated and characterized (Supplementary Figs. 8 and 13). The observed $T_g$ of 1 °C is significantly lower than that of polymer **P1**, which can be attributed to the longer alkyl chains and thus lower relative amount of urethane moieties. In the case of monomer **15**, the molecular weight is limited (entry 5, Table 4), which could be related to an increased stiffness. The molecular weight was not sufficiently high for a successful precipitation; thus, the polymer could not be characterized for molecular and thermal analysis. Monomer **19** could also be successfully used for the synthesis of NIPUs. When using dithiol **20** (entry 8, Table 4), the polymer **P8** with the highest molecular weight within this work of 31.2 kDa was obtained after precipitation. Its $T_g$ of 23 °C is higher than that of **P6**, where the same amine was used for the carbonate opening.

The higher $T_g$ can be attributed to increased hydrogen bonding due to the presence of two urethane moieties per repeating unit. Structural characterization of **P8** is found in Supplementary Figs. 9 and 14.

For a variation of the dithiol, the urethane monomer **13** was chosen as starting point. The use of dithiol **21** with a shorter chain length led to a polymer with lower molecular weight (entry 2, Table 4), which might be attributed to a less favorable structure in which the terpene units are relatively close. The resulting polymer **P2** was purified and characterized (Supplementary Figs. 7 and 12), showing a $T_g$ of 16 °C that is similar to that of **P1**. The renewable dithiol **22** in combination with monomer **13** did not yield a polymer with high molecular weights (entry 3, Table 4), supporting the previous assumption of close terpene moieties hampering the formation of longer polymer chains. For this reason, dithiol **22** was tested for a polymerization with monomers **18** and **19**, which contain one and two additional decenyl spacers, respectively. Indeed, already the use of monomer **18** (entry 7, Table 4) with an elongated chain length compared to monomer **13** (entry 3, Table 4) led to higher molecular weight, yet not high enough for a precipitation of the polymer **P7**. Extending this to monomer **19** with an additional $C_{10}$ chain led to the formation of NIPU **P9** that could be precipitated and characterized (see Supplementary Figs. 10 and 15). As in the NMR spectrum of **P9** the double bond signals are still visible as end groups, their integration can be used to calculate the molecular weight of **P9** (see Supplementary Fig. 10 and Supplementary Eq. (1)). The calculated molecular weight of 11.8 kDa is in a similar range as the value from SEC measurements (see Table 4, entry 9). The observed $T_g$ of 29 °C

is higher than that of **P8** with a linear dithiol, corresponding to a higher stiffness of the terpene unit.

## Discussion

This work showed the application of thiourea catalysis for the functionalization of terpene-based carbonates towards urethane building blocks. The presence of a thiourea catalyst significantly improved the opening of the endocyclic carbonate groups by allylamine, whereas no activation was necessary in the case of the exocyclic carbonate structures. This enabled the access to AA monomers for the synthesis of linear NIPUs as potential application in polymer synthesis. By thiol-ene polyaddition with dithiols, NIPUs with molecular weights of up to 31 kDa were obtained, strongly depending on the structure of the respective monomers. By elongating the carbon chains within the urethane monomers, it was possible to achieve higher molecular weights and further implement a renewable dithiol from limonene. The $T_g$ values, ranging from 1 to 29 °C, are slightly higher than those of literature-described terpene-containing NIPUs of similar molecular weight[37,59,67], which can be attributed to additional OH groups[59,67] or higher terpene content, respectively[37].

This approach complements previous strategies of introducing urethane moieties into polymers via thiol-ene reaction[59,68]. It should be noted that the obtained materials contain additional thioether linkages as well as hydroxy groups in contrast to industrially used PUs. However, other works also include thioether linkages, e.g. for self-blowing NIPU foams[69,70], showing the potential of such new structures. Further, this work brings forward the use of thiourea catalysis for NIPU production[71], as potential strategy to activate more hindered cyclic carbonates. Although several examples have shown the potential of implementing terpene structures into polyurethanes[22,36–38,67,72,73], their number remains limited.

## Methods

Materials and method procedures are provided in Supplementary Methods. Synthetic procedures and characterization details are provided in the Supplementary Notes 1–3.

For NMR spectra and GC-FID chromatograms of isolated compounds, see Supplementary Figs. 17–75.

## Data availability

The authors declare that the data supporting the findings of this study are available within the article and Supplementary Information. For experimental details and compound characterization data, see Supplementary Information. All other data are available from the corresponding author on reasonable request.

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

## Acknowledgements

The authors want to acknowledge Michelle Karsten and Elsa Brudy for synthetic support and the analytical department of the Institute of Organic Chemistry (IOC) at KIT and Dr. Andreas Rapp, Despina Savvidou, Tanja Ohmer-Scherrer, and Lara Hirsch for their support with the NMR and ESI measurements.

## Author contributions

F.C.M.S. and M.A.R.M. conceived and designed the project. F.C.M.S. is the lead author of the manuscript and carried out and directed the experiments and synthesis of the compounds. M.A.R.M. directed the work and revised the manuscript.

## Funding

## Competing interests

The authors declare no competing interests.
