## [Peer review file · Communications Chemistry]

Non-isocyanate polyurethanes synthesized from terpenes using thiourea organocatalysis and thiol-ene-chemistryReviewers' comments:

Reviewer #1 (Remarks to the Author):

- The authors describe an interesting study about 'Non-Isocyanate Polyurethanes from Terpenes Using Thiourea Organocatalysis and Thiol-Ene-Reaction'. The five-membered cyclic carbonates based on the terpenes limonene and carvone (some of them are already known from literature), their ring-opened products (with allyl amine using thiourea organocatalysts) resulting in renewable AA monomers, and the resulting NIPU polymers are the main convincing claims of this paper. They have high interest to others in the community and the wider fields and also the potential to influence thinking in the fields. The following minor revisions should be made prior to its publication:
- Figure 5: Please show the graph over a larger retention time period and briefly comment about the term 'system peak' more detailed in this context.
- Discussion: Please compare these new NIPUs with some established NIPUs a bit more detailed.
- A photograph of some selected NIPUs should be shown in order to get an impression of their appearance.
- SI: The further size-exclusion chromatography/ gel permeation chromatography (SEC/GPC) traces of the NIPUs polymers P1-P9 are shown here, but should be discussed a bit more detailed also for the oligo-SEC samples.

Reviewer #2 (Remarks to the Author):

The manuscript by Scheelje et al. reports on the synthesis of urethane moiety containing monomers originating from terpenes (limonene) via cyclic carbonates. Corresponding polyurethane were synthesized by thiol-ene polymerization. The data presented are sound and mark an improvement to existing knowledge. The topic of the manuscript matches the scope of Chemical Communications and could be published there.

The weak part of the manuscript that no clear conclusion are provided in order to support the readers in valuing the authors' accomplishment and the document the progress. The authors are highly recommended to elaborate meaningful conclusions.

Reviewer #3 (Remarks to the Author):

The paper presents a study on the synthesis of non-isocyanate polyurethanes (NIPUs) from terpenes by thiourea organocatalysis and UV triggered thiol-ene reaction. In this paper, synthesis of carbonate monomers, optimized conditions of (endo-, exo-) ring-opening & thiol-ene reactions, and variation of dithiols & dienes towards NIPUs are thoroughly researched. This paper has a significant implication for the research fields of utilizing renewable biomass for high-value utilization, and synthesizing NIPU under non-toxic or low toxicity conditions. Minor revision is recommended, and my specific comments are shown below:

1. The reference to "Monomer 16" in line 164 is unclear and has not been mentioned in the nearby context.
2. Line 187-188, "Further, the initial choice of 70 °C proved 187 to be the most suited for the investigated reaction (see entries 4-5, Table 1)." However, the results was not included in the Table
3. The bolded data in entry 2, 3, 4, and 5 of Table 1, corresponding to the discussion of the optimal ring-opening conditions in line 213 and 214. It is better to compare the bolded data in the context near Table 1.
4. It is recommended to use "in the case of" instead of "in case of" in lines 80, 231, 263, 272, 314, 367, and 401.
5. The explanation of lines 257-258 "This reports a drawback of the bifunctional monomer 15 when compared to monomer 13" is unclear.
6. Line 386-387, "Indeed, already the use of monomer 18 with an elongated chain length compared to

monomer 13 led to higher molecular weight (entry 7, Table 4), yet not high enough for a precipitation of 387 the polymer P7." There is not "Monomer 13" in the footnote.

7. Supplementary Figure S10 shows that the double bond group has two arrows, while Formula 1 only uses one arrow for calculation.

8. There is limited experimental data in the abstract, and most sentences regarding background information may be slightly deleted.

Answer to reviewers' comments:

black: original reviewer comments

green: response from the authors

We would like to thank all reviewers for their time and efforts to review our manuscript, which definitely contributed to an improvement of the manuscript. In addition to changes made in response to the reviewer's comments, minor mistakes were corrected. All changes within the manuscript were highlighted.

Reviewer #1 (Remarks to the Author):

- The authors describe an interesting study about 'Non-Isocyanate Polyurethanes from Terpenes Using Thiourea Organocatalysis and Thiol-Ene-Reaction'. The five-membered cyclic carbonates based on the terpenes limonene and carvone (some of them are already known from literature), their ring-opened products (with allyl amine using thiourea organocatalysts) resulting in renewable AA monomers, and the resulting NIPU polymers are the main convincing claims of this paper. They have high interest to others in the community and the wider fields and also the potential to influence thinking in the fields. The following minor revisions should be made prior to its publication:

- Figure 5: Please show the graph over a larger retention time period and briefly comment about the term 'system peak' more detailed in this context.

Answer: Thank you for this remark. We show a zoom into the chromatogram in Figure 5 to be able to easier distinguish the two relevant signals. We decided to leave it as such, as it will be easier for the reader to take the main message, but we are happy supply you the full chromatogram here (which will be available to the public due to transparent peer review):

The signals after 20 minutes retention time are termed system peaks, as they appear in all measurements, including the blank (solvent) samples measured in advance. To demonstrate this, also show the blank THF measurement below that was obtained directly before the obtaining the chromatogram above, also displaying the respective signals:

- Discussion: Please compare these new NIPUs with some established NIPUs a bit more detailed.

Answer: We implemented this helpful comment in the written conclusion. The molecular weights and T_g values were compared to literature-described polyurethanes containing terpene structures.

- A photograph of some selected NIPUs should be shown in order to get an impression of their appearance.

Answer: Thank you for this comment. Due to the synthesis of linear NIPUs being mostly a proof of concept within this work, which focused on catalytic monomer synthesis, most polymers were synthesized in small amounts (see Supplementary Information, section 2.3). To address your question, we made pictures of the precipitated polymers **P1** (left), **P6** (middle) and **P9** (right) with T_g values of 18 °C, 1 °C and 29 °C, respectively. We are of the opinion that adding these pictures to the manuscript is not of large value to the reader (as above, these pictures will however be public due to transparent peer review).

- SI: The further size-exclusion chromatography/ gel permeation chromatography (SEC/GPC) traces of the NIPUs polymers P1-P9 are shown here, but should be discussed a bit more detailed also for the oligo-SEC samples.

Answer: Thank you for the suggestion. We added a respective sentence to the main text as follows:
 "SEC of the obtained polymeric materials was performed (chromatograms are shown in Supplementary Figures 3-5) to determine M_n and \mathcal{D} (see Table 4), revealing mostly high molecular weights and dispersities close to 2, as expected for a step-growth polymerization."

Reviewer #2 (Remarks to the Author):

The manuscript by Scheelje et al. reports on the synthesis of urethane moiety containing monomers originating from terpenes (limonene) via cyclic carbonates. Corresponding polyurethane were synthesized by thiol-ene polymerization. The data presented are sound and mark an improvement to existing knowledge. The topic of the manuscript matches the scope of Chemical Communications and could be published there.

The weak part of the manuscript that no clear conclusion are provided in order to support the readers in valuing the authors's accomplishment and the document the progress. The authors are highly recommended to elaborate meaningful conclusions.

Answer: Thank you for this remark. We extended the written conclusions and implemented a comparison of the approach to previous works on terpene-based polyurethanes, further contextualizing the obtained results within the current research field on the synthesis of more sustainable nitrogen-containing polymers.

Reviewer #3 (Remarks to the Author):

The paper presents a study on the synthesis of non-isocyanate polyurethanes (NIPUs) from terpenes by thiourea organocatalysis and UV triggered thiol-ene reaction. In this paper, synthesis of carbonate monomers, optimized conditions of (endo-, exo-) ring-opening & thiol-ene reactions, and variation of dithiols & dienes towards NIPUs are thoroughly researched. This paper has a significant implication for the research fields of utilizing renewable biomass for high-value utilization, and synthesizing NIPU under non-toxic or low toxicity conditions. Minor revision is recommended, and my specific comments are shown below:

1. The reference to "Monomer 16" in line 164 is unclear and has not been mentioned in the nearby context.

Answer: Indeed, this was a wrong number, thank you for pointing this out, we corrected it to "monomer 13".

2. Line 187-188, "Further, the initial choice of 70 °C proved 187 to be the most suited for the investigated reaction (see entries 4-5, Table 1)." However, the results was not included in the Table

Answer: We apologize for the description being imprecise, we meant to say that the original conditions from entry 1, Table 1 were superior to the reactions with varied temperature (entries 4-5, Table 1). We clarified this in the manuscript, it now reads as follows:

"Further, the initial choice of 70 °C (entry 1, Table 1) proved to be the most suited for the investigated reaction if compared to temperatures of 60 and 80 °C (see entries 4-5, Table 1)."

3. The bolded data in entry 2, 3, 4, and 5 of Table 1, corresponding to the discussion of the optimal ring-opening conditions in line 213 and 214. It is better to compare the bolded data in the context near Table 1.

Answer: Also in this case, we are sorry for the unclear wording. The optimum conditions are actually found in entry 1, Table 1, and the bolded data was to highlight the changes that were made during the screening. We agree that this might not be confusing, especially with the aim to understand the table without having to read the whole text. Therefore, we eliminated the bold font, so the variation of reaction conditions can be read from the text.

4. It is recommended to use “in the case of” instead of “in case of” in lines 80, 231, 263, 272, 314, 367, and 401.

Answer: Thank you for pointing this out, we changed it accordingly.

5. The explanation of lines 257-258 “This reports a drawback of the bifunctional monomer 15 when compared to monomer 13” is unclear.

Answer: Thank you, we added some further discussion to clarify this, the changed sentence reads as follows:

"The occurring of this structurally similar intermediate, which is not straightforward to isolate from the product, represents a drawback of the bifunctional monomer 15 when compared to monomer 13."

6. Line 386-387, “Indeed, already the use of monomer 18 with an elongated chain length compared to monomer 13 led to higher molecular weight (entry 7, Table 4), yet not high enough for a precipitation of 387 the polymer P7.” There is not “Monomer 13” in the footnote.

Answer: Thank you for the indication, we noticed that the brackets were not placed optimally. As the aim was to compare entries 3 and 7 of Table 4, corresponding to the use of either monomer 13 (entry 3) or monomer 18 (entry 7), it makes more sense to write this in separate brackets, which we changed in the text accordingly as follows:

"Indeed, already the use of monomer 18 (entry 7, Table 4) with an elongated chain length compared to monomer 13 (entry 3, Table 4) led to higher molecular weight, yet not high enough for a precipitation of the polymer P7."

7. Supplementary Figure S10 shows that the double bond group has two arrows, while Formula 1 only uses one arrow for calculation.

Answer: Thank you, indeed only one of the signals is used for the calculation. The double bonds of the monomers show two distinct signals (see Figure S58), but one of them overlaps with the OH/ NH signal in Figure S10. Therefore, only the signal from 5.90 to 5.73 ppm was taken into account for the calculation of the molecular weight. In the monomer, its integral value would be 2, which was considered in the formula. In principle, the same calculation could be done with the second double bond signal to verify the result, however, the signal overlap made this not feasible. Still, we pointed out the signal in the Figure as it is clearly visible and its presence confirms the assumption of detected end group signals.

8. There is limited experimental data in the abstract, and most sentences regarding background information may be slightly deleted.

Answer: The inclusion of experimental data in the abstract is indeed worthwhile, therefore we included the T_g range of the polymers and further added some more information about the monomer scope. Regarding the background information, we agree that it is not absolutely necessary to include an introductory phrase in the beginning of the abstract, however, the Style and Formatting Guide of the Communications Journals explicitly states that the abstract “should include the background and context of the work”.